# Discovering individual-specific gait signatures from data-driven models of neuromechanical dynamics

**Taniel S. Winner** [1]*, **Michael C. Rosenberg**[1], **Kanishk Jain**[2], **Trisha M. Kesar**[3], **Lena H. Ting**[1,3], **Gordon J. Berman**[4]*

1 W.H. Coulter Dept. Biomedical Engineering, Emory University and Georgia Institute of Technology, Atlanta, Georgia, United States of America, 2 Department of Physics, Emory University, Atlanta, Georgia, United States of America, 3 Department of Rehabilitation Medicine, Division of Physical Therapy, Emory University, Atlanta, Georgia, United States of America, 4 Department of Biology, Emory University, Atlanta, Georgia, United States of America

* twinner@emory.edu (TSW); gordon.berman@emory.edu (GJB)

## Abstract

Locomotion results from the interactions of highly nonlinear neural and biomechanical dynamics. Accordingly, understanding gait dynamics across behavioral conditions and individuals based on detailed modeling of the underlying neuromechanical system has proven difficult. Here, we develop a data-driven and generative modeling approach that recapitulates the dynamical features of gait behaviors to enable more holistic and interpretable characterizations and comparisons of gait dynamics. Specifically, gait dynamics of multiple individuals are predicted by a dynamical model that defines a common, low-dimensional, latent space to compare group and individual differences. We find that highly individualized dynamics–i.e., gait signatures–for healthy older adults and stroke survivors during treadmill walking are conserved across gait speed. Gait signatures further reveal individual differences in gait dynamics, even in individuals with similar functional deficits. Moreover, components of gait signatures can be biomechanically interpreted and manipulated to reveal their relationships to observed spatiotemporal joint coordination patterns. Lastly, the gait dynamics model can predict the time evolution of joint coordination based on an initial static posture. Our gait signatures framework thus provides a generalizable, holistic method for characterizing and predicting cyclic, dynamical motor behavior that may generalize across species, pathologies, and gait perturbations.

## Author summary

In this manuscript, we introduce a novel, machine learning-based framework for quantifying, characterizing, and modifying the underlying neuromechanical dynamics that drive unique gait patterns. Standard methods for evaluating movement typically focus on extracting discrete gait variables ignoring the complex inter-limb and inter-joint spatiotemporal dependencies that occur during gait. Popular physiologically realistic modeling approaches encode these spatiotemporal dependencies but are too complex to

**Data Availability Statement:** To ensure rigor, reproducibility, and promote open science, all software is shared under a GNU GPL 3.0 license

and on GitHub. Links to Google Colab notebooks enable our software to be run on the cloud for users without computational resources. The gait signature code developed and used in this paper has been deposited at GitHub: https://github.com/bermanlabemory/gait_signatures.The RNN model and code was developed in Python programming language using built-in Python-based libraries such as Keras, Pandas, and NumPy. We revised the Phaser algorithm (https://github.com/sheim/phaser) to estimate phase for our kinematic time trajectories in the development of our phase averaged dynamics per trial. Shareable Jupyter Notebooks were developed on the Google Colab platform. The data analysis of the generated gait signatures was conducted in MATLAB 2022a (MathWorks).

**Funding:** This work was supported by the Alfred P. Sloan Foundation's Minority Ph.D. (MPHD) program: G-2019-11435, NSF GRFP 1937971 and NICHD F31HD107968 for TSW, NICHD F32HD108927 for MCR, HFSP RGY0076/2018 and NSF Physics of Living Systems Student Research Network (PHY-1806833) for KJ, NICHD R01HD095975 for TMK, the McCamish Foundation for TSW, MCR and LHT, NSF CMMI 1762211 / 1761679 for LHT, the Simons-Emory International Consortium on Motor Control (Simons Foundation, 707102) for TSW, KJ and GJB and an Emory University Nexus/Synergy II Grant for all authors. The funders had no role in study design, data collection and analysis, decision to publish, or preparation of the manuscript.

**Competing interests:** The authors have declared that no competing interests exist.

characterize individual differences in the factors driving unique gait patterns or disorders. To circumvent these modeling complications, we develop a phenomenological model of gait that enables more holistic and interpretable characterizations of gait, encoding these complex spatiotemporal dependencies between humans' joint angles arising from joint neural and biomechanical constraints. Our coined 'gait signature' framework provides a path towards understanding the neuromechanics of locomotion. This framework has potential utility for clinical researchers prescribing individualized therapies for pathologies or biomechanists interested in animal locomotion or other periodic movements assessed across different pathologies, neural perturbations, and or conditions.

## Introduction

Locomotion is a ubiquitous, complex, and dynamic behavior that is essential for survival. Using cyclic patterns of joint angles, inter-limb and inter-joint coordination, animals effectively move through their environments: walking, running, trotting, swimming, flying, and crawling. Even within species and types of locomotion, variations in locomotor patterns often occur across behavioral contexts, groups, and individuals. Thus, although locomotor patterns can appear highly stereotyped, considerable inter- and intra-individual variability exists. Studies of locomotor behaviors have shown systematic differences in movement patterns based on a wide range of neural [1–4] and biomechanical perturbations [5–8] environmental challenges [9,10], psychological state [11,12], social status [13,14], injury [15–17], and disease [4,18–23]. Furthermore, locomotor impairments can arise from a wide range of physiological and neurological changes, from the subtle changes that may be indicators of progressive disorders (e.g., aging, cognitive impairments) to profound impairments with brain injury (e.g., stroke, spinal cord injury) that can severely limit locomotor function. Although locomotor deficits are often subjectively visible to a human observer, objectively characterizing and understanding sometimes subtle yet important differences in locomotion from a scientific and mechanistic standpoint has been challenging [24–26]. For example, kinematic movement patterns (the continuous motion of joint angles over time) have been collected across a wide range of locomotor modes and species but revealing individual-specific differences in kinematics remains difficult. One barrier to progress is that interpreting individual differences in kinematics without an underlying dynamical model is challenging, as kinematics are the result of the complex neuromechanical dynamics that drive the spatiotemporal dependencies of joint kinematics over time. Thus, capturing these underlying gait dynamics is likely essential for interpreting differences in gait and movement across conditions and individuals.

Traditionally, gait dynamics are modeled using physiologically detailed neuromechanical equations, however making predictive models using this approach has often proved challenging [27–29]. Partially, this difficulty arises because in order to understand the dynamics underlying gait, we also need to understand how neural feedback and control shape these dynamics. While many models (e.g., musculoskeletal models) that use principles like optimal control can generate simulations of unimpaired gait, as well as changes in gait due to altered biomechanical or neural constraints, they often fail to predict changes in gait kinematics following neurological injury [28] or more subtle perturbations [30,31]. Progress in the physiological modeling of locomotor circuitry in the spinal cord and brainstem demonstrates the role of neural circuits in gait dynamics. However, these models typically rely on simplified [32,33] biomechanical properties and cannot yet predict the deficits in gait specific to an individual [25,34–36]. More importantly, if a hyper-realistic model of the neural and biomechanical

system did exist, the relationships between the high-dimensional parameters and actual movement patterns would not likely be unique, as many parameters would not be identifiable, even given massive amounts of data, as many different parameter choices could lead to the same biomechanical output [37,38]. This non-identifiability limits the predictive power and generalizability of these models to other interventions and conditions outside of limited contexts, suggesting a need for a more holistic approach.

Despite these challenges, rich individual-specific information exists in gait data. For instance, through observation of movement, the human brain can perceive many socially salient features of an individual's gait, suggesting that it should be possible to infer aspects of gait dynamics from kinematic data. As an example, humans can derive a host of information about individuals from movement patterns, including gender [39], body size [40], sexual orientation [41], emotion [42], individual differences in dancing [43], perceived affective states [44] and underlying intention [45]. Furthermore, judgements based on how individuals move can drive decisions such as partner desirability or attractiveness [46] diagnosis [47,48], and treatment planning [49,50].

Despite the rapid advent of technologies providing kinematic measurements through a wide range of techniques, from videos to wearable sensors, we are still limited in how kinematic data can help interpret individual differences in gait [51,52]. Current approaches to comparing biomechanical features or kinematic trajectories quantify between-group differences or inter-individual similarity but lack sufficient sensitivity to reveal interpretable differences in individuals' gaits [53–55]. Inter-joint coordination differs across individuals, as muscular coordination patterns vary across a variety of motor skills and deficits in individual-specific ways. Indeed, metrics of muscle coordination in children with cerebral palsy are consistent with clinician judgements of motor control complexity that predict intervention outcomes [56]. Recently, supervised machine learning methods have been used to classify differences in a large sets of gait kinematics that were labeled by groups or individuals [55,57]. However, these approaches have not modeled the underlying gait dynamics, nor can they discover subtle differences in gait that are not labeled a priori.

Here we develop a data-driven framework for modeling gait dynamics that represents multiple individuals in the same latent space. This latent space reveals individual- and group-level differences in the neuromechanical dynamics of gait. We used kinematic data from multiple healthy and neurologically impaired individuals, each walking at six different speeds, to train a recurrent neural network (RNN) that learns gait dynamics. This phenomenological approach infers complex spatiotemporal dynamics and enables future kinematic predictions to be made based on current and prior kinematic postures. Once trained, differences in gait dynamics across groups, individuals, and walking speed were projected onto a common, low-dimensional latent space of the model parameters. The stride-averaged representation of gait dynamics in the latent space constitutes a "gait signature" that we use to characterize differences across individuals, groups, gait speed, and impairment severity. To demonstrate the generalizability of gait dynamics, we show that interpolating gait signatures to predict gait kinematics at new walking speeds is more accurate than interpolating the kinematics themselves in healthy individuals. Further, we show that the low-dimensional basis functions we discovered have biomechanical interpretability in terms of the inter- and intra-limb coordination patterns that they generate. The dynamical projections onto each basis function for each trial can be independently driven through the trained gait dynamics model to reconstruct the kinematics associated with that specific basis function. We generated illustrations of the reconstructed joint angle kinematics to visualize and infer what aspects of gait coordination each subcomponent influences. These subcomponents of gait coordination can be manipulated independently (i.e., gait sculpting) to infer the relationships between specific underlying dynamical components

and their corresponding kinematic phenotypes and to identify what specific gait rehabilitation strategies are likely required for individuals. Finally, our gait dynamics model is generative; it can predict individual-specific time evolution of kinematics from an initial arbitrary posture (self-driving) once the network is primed with several gait cycles of the individual's kinematic data. This study establishes a new data-driven framework to quantitatively interpret individual-specific differences in gait dynamics with the potential to enable discovery in a wide range of gait coordination deficits, contexts and interventions in humans and other animals.

## Results

### Gait signatures: A low-dimensional representation of gait dynamics

We used motion capture to collect sagittal-plane kinematic data that consisted of 15 seconds of continuous gait kinematics from bilateral, hip, knee, and ankle joints from 5 able-bodied (AB) participants and 7 stroke survivors (> 6 months post-stroke, gait speeds 0.1 to 0.8 m/s) walking on a treadmill at a range of six different speeds each. Taking inspiration from neural network models that capture neural dynamics [58–60] and biological systems, we implemented a recurrent neural network (RNN) model to capture the dynamical properties of gait. Our model input parameters only include kinematic data and do not include anthropometric information or clinical characteristics and do not account for differences in joint kinematics due to neural versus biomechanical constraints.

### Developing the recurrent neural network (RNN) architecture and training the model

The gait dynamics model was developed in Python using common Python libraries, including TensorFlow, Keras, Pandas, and NumPy. We developed our code in Google Colab to facilitate open-source sharing of our dynamic framework, which can be found here: https://github.com/bermanlabemory/gait_signatures. The model architecture was selected based on two criteria: 1) minimizing model training and validation loss during model fitting, and 2) maximizing the similarity of short-time (single stride) and long-time (multiple strides) self-driven model predictions (termed: *gait signature alignment*) post model training (S1 Fig). By implementing these two model selection criteria we ensure 1) a high goodness-of-fit (model that best represents the underlying dynamics across all participants and gait speeds) and 2) the model is capable of predicting the time-evolution of gait (encode gait dynamics). We evaluated these criteria against alternative models by varying 2 hyperparameters (number of LSTM units and the look-back time, see Methods). The selected model architecture is a sequence-to-sequence RNN [61] consisting of an input layer, a hidden layer of 512 LSTM units, and an output layer. The RNN learns a map from time-series kinematic input data (0 to T-1) to kinematics one time-step in the future (1 to T) for all training trials (Fig 1A). The model was trained using the 'mean squared error' (mse) loss function until training and validation error converged and stabilized around the same point ($< 0.03$ degrees$^2$). Thus, the model successfully learns the underlying dynamics of gait (S2 Fig). The model's internal states capture trial-specific dynamics predicting the time evolution of joint kinematics; activation coefficients (H) and memory cell states (C) and are tuned based on kinematic inputs. Kinematic data was input in multivariate format, not concatenated [62,63]. In brief, our RNN model was designed to capture short and long-term gait dependencies in time [64,65] as well as inter-and intra-limb coordination over time, uncovering features of gait that were not previously targeted or used in gait analysis. To verify whether our model was generalizable, we conducted leave-one-out cross validation, where 12 different models were trained leaving a single individual's 6 trials on each model run (S3 Fig).

**Fig 1. Pipeline figure outlining the steps to generating individual-specific gait signatures.** Continuous, multi-joint kinematics from multiple individuals are fed into the RNN model as input data and the model is trained sequence-to-sequence to predict one-step time shifted output kinematics. High dimensional internal parameter (H and C) time traces per individual are extracted and principal component analysis was applied to reduce the dimensionality of the data to form individual gait signatures (A). 3D time trace visualizations of 3 representative individuals (able-bodied (blue), high-functioning (red), low-functioning stroke (orange)) of the 1st 3 dominant principal component contributions (B, left). 3D projections of the 6-D gait signatures using multi-dimensional scaling (MDS) reveal different gait dynamics amongst the three gait groups: able-bodied (blue), high-functioning (red) and low-functioning (orange) stroke survivors (B, right). The size of the circles represents the individual's trial speed (i.e., the smallest circles represent an individual's slowest gait speed, and the size of the circles increase with gait speed).

Stroke-survivors are known for having neurological impairments that result in heterogeneous gait dysfunction that are not fully understood; thus, we anticipate that our gait dynamics model will capture and shed light on these individual-specific deficits in gait coordination, identify similar coordination strategies or deficits amongst our stroke cohort, and allow us to compare these different gait dysfunctions to the able-bodied 'normative' gait (controls).

## Generating gait signatures

To generate gait signatures, kinematic trajectories from each walking speed trial across participants were fed as input into the trained neural network and the corresponding internal states (H and C parameters, see above) were extracted (Fig 1A). The internal activations prescribe the spatial and temporal dependencies generating the input kinematics. The resulting time-series of 1024 internal states (512 H, 512 C parameters) were reduced in dimension using Principal Components Analysis (PCA) and phase averaged [66]. Phase averaging is applicable here, as the underlying gait dynamics are periodic, and the translation from time to a phase between 0 and $2\pi$ allows us to describe all internal state dynamics in a speed-independent manner.

The first 6 Principal Components (PCs) explain ~72% of the variance in gait dynamics (S4 Fig), allowing us to focus on these modes for our visualization and analysis. The time-varying contributions of the first 3 dominant PCs were plotted in 3D for 3 representative individuals—able-bodied adults, high-functioning stroke (self-selected (SS) walking speed > 0.4m/s) and low-functioning stroke (SS speed < 0.4m/s)—highlighting that the gait dynamics between all 3 individuals are different (Fig 1B, left). The gait dynamics of the high-functioning stroke survivor (red), while spatially closer to the able-bodied individual (blue) than the low-functioning stroke survivor (orange), show observable differences in its dynamical trajectory between to the two individuals. To determine whether some structure exists amongst the three different subject groups, all the 6-dimensional gait signatures were projected onto a 3D map using Multidimensional scaling (MDS) [67] to visualize relative distances between all gait signatures

(Fig 1B, right). The locations of the 3 MDS projections of the 3 representative individuals are not arbitrary, as they belong to clusters of gait signatures of the same gait group. Thus, gait signatures preserve key clinically relevant features of the underlying gait dynamics, independent of the individual or speed.

## Gait signatures reveal that individual-specific differences in dynamics are favored in the gait representation, over differences in gait speed

Gait signatures of individuals' 6 speed trials within both cohorts (healthy and stroke) are tightly grouped together. Gait signatures represent individual-specific dynamics; the unimpaired cohort exhibit a stereotyped low-dimensional structure across individuals in the able-bodied cohort (Fig 2Ai, left) vs. the impaired cohort, which display much more variable (i.e., highly individualized) low-dimensional representations (Fig 2Ai, right). Because the data are phase averaged over the gait cycle, we demonstrate that gait signature trajectories are well-aligned with the four gait phases (leg 1 swing, leg 1 stance, leg 2 swing, leg 2 stance), enabling phase-specific comparisons of differences in gait dynamics. The unimpaired group showed similar structure across the four gait events (Fig 2Aii, left), whereas there was much more variability within the impaired group (Fig 2Aii, right), revealing individual-specific differences within and across distinct parts of the gait cycle. The similarity between gait signatures was

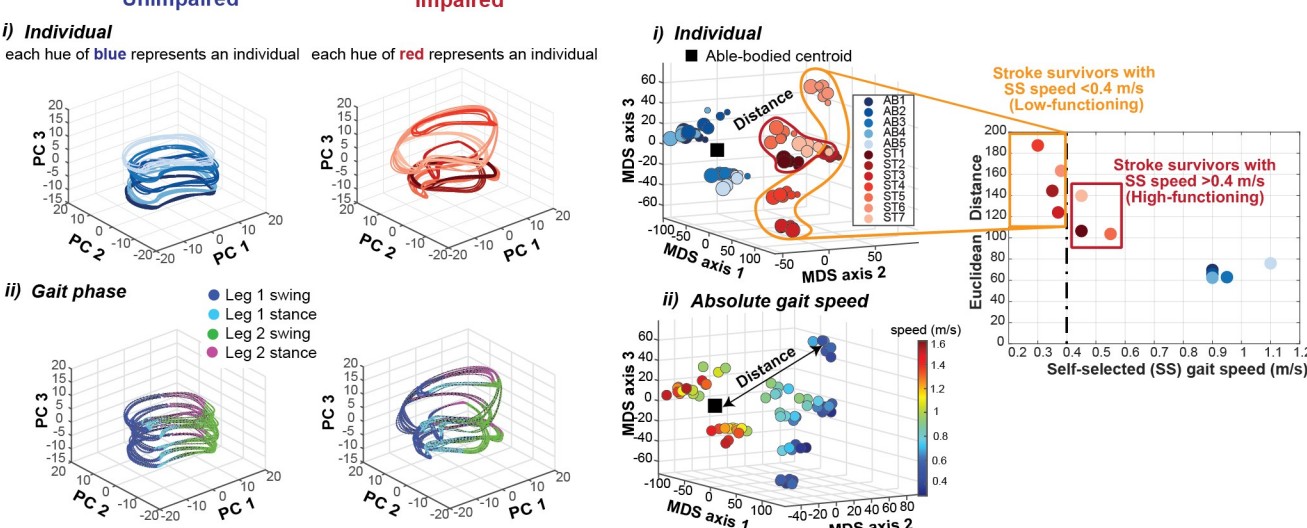

**Fig 2. Gait signatures reveal highly individualized low dimensional representations of gait dynamics irrespective of absolute gait speed.** A) 3D unimpaired (left) and impaired (right) gait signatures colored by i) individual and ii) gait phase. Gait signatures are grouped together according to individuals within both cohorts (same hues of blue cluster together for unimpaired (i, left) and similarly the same hues of red cluster in the impaired cohort (i, right)). In our convention the right leg of all unimpaired individuals was assigned to be the paretic leg and left leg the non-paretic leg. Impaired individuals can have either left or right leg paresis. Unimpaired gait signatures reveal a similar looped structure across the four gait phases that occur during a gait cycle (leg 1 swing, leg 1 stance, leg 2 swing, leg 2 stance) (ii, left) whereas impaired signatures showed individual-specific differences across the four phases and were more variable (ii, right). B) 3D multidimensional scaling applied to all gait signatures shows the pronounced separation between unimpaired (blue hues in left section of map) and impaired (red hues in right section of map) gait dynamics (i). Impaired signatures (red hues) are located further away from the centroid of all unimpaired gait signatures (black square), indicating that they are less dynamically similar to the unimpaired individuals. The smallest circles represent an individual's self-selected walking speed trial and larger circles correspond to the faster speed trials. Low-functioning stroke survivors (encapsulated in orange; based on self-selected gait speed < 0.4m/s) are located furthest away (largest Euclidean distances) from the unimpaired centroid (i). Gait speed does not appear to strongly influence the differences in dynamics between individuals as similar speed gait signatures are in different regions of the gait map (ii). Particularly, gait speed does not explain the heterogeneity in low-functioning stroke survivors' gait dynamics.

computed and visualized in a dimensionally reduced gait map space using MDS and colored according to the different individuals in the dataset (Fig 2Bi). The unimpaired group form a cluster in the gait map, showing that individuals in the unimpaired group are distinct from the impaired group. Stroke-survivors occupy distinct positions from other impaired individuals' sub-clusters in the gait space that highlight the well-established but poorly understood heterogeneity in gait deficits in the stroke cohort. Furthermore, individual-specific gait signatures change slightly as individuals walk faster than their self-selected pace (Fig 2Bi). However, these within-subject speed-induced changes are much smaller than between-individual difference in gait signatures. The gait signatures of the individuals belonging to the able-bodied, high-functioning, and low functioning stroke survivor cohorts show 3x, 4x and 7x larger distances between individuals in the group versus within each individual 6 speed trials, respectively. We calculated the Euclidean distance between individuals' self-selected speed trial gait signature and the calculated able-bodied centroid (Fig 2Bi, black square) and the results shown on the plot to the right reveal that low-functioning stroke survivors (characterized based on the clinical definition of having a self-selected walking speed of <0.4 m/s) are further away from the able-bodied cluster than the high-functioning stroke survivors. Note that no information from a clustering algorithm was used to characterize low- vs high-functioning individuals. The encircled low-functioning stroke survivors (Fig 2Bi, orange enclosure) were labeled post-hoc to demonstrate the lack of a single cluster characterizing low- vs high-functioning stroke survivors. Showing the validity of our approach, low-functioning stroke survivors are less dynamically similar to AB than higher functioning stroke survivors.

Gait speed does not appear to strongly influence the differences in dynamics between individuals' gaits (although the range of gait speeds for each participant may not have been wide enough to elicit major differences in their overall dynamics). It is worth noting that the walking speeds in our post-stroke cohort spanned the full speed range of each participant's safe walking capacity, whereas this was not the case for our able-bodied cohort. Overall, as expected, the unimpaired group walked at faster speeds than the impaired group (Fig 2Bii). Individuals in the able-bodied cluster walk at a range of different speeds, but individual gait signatures still cluster tightly together. Despite the able-bodied dynamics being similar, there still exists inherent variability in their gait dynamics that may be explained by factors such as prior exercise and sports-training history, injury, disease, etc. Post-stroke individuals who walk at similar slower speeds, however, maintain their own distinct individualized groupings. Thus, individuals' characteristic gait signatures were preserved across their range of walking speeds and were not grouped based on absolute walking speed. For example, several clinically similar post-stroke individuals (similar overground walking speed and Fugl-Meyer score [68]) have very different gait signatures that remain recognizable across a range of gait speeds (Fig 2). Although the low-functioning individuals in our sample are more dispersed than high-functioning individuals, we expect that the spaces between individuals represent a continuum of gait dynamics that would be filled given a larger sample size.

Furthermore, when used to distinguish between gait groups and identify individuals, gait signatures perform similarly to using a set of 26 commonly used discrete variables (S5 Fig). Discrete variables are already sufficient to classify between able-bodied and stroke gaits, with numerous studies identifying key variables that map to function/impairment [69–71]. With Gait signatures, we achieve the same level of classification without needing to hand-pick discrete variables or to use force plates or inverse-dynamics analyses that would require more equipment, computation, and subject-specific anthropometry for each observation. Gait signatures also perform better than continuous kinematics and joint velocities at these same discrimination tasks (S5 Fig). These results serve as a positive control, as researchers previously could distinguish gait groups by building a classifier based on important subjectively selected

discrete variables. Here, we have created a dynamical representation that can distinguish groups with similar accuracy. It is not surprising that the continuous kinematics performed worse than the RNN gait signatures (which were developed from these very same data), as the RNN model used the data to encode important time-varying changes in the kinematics, allowing for more information to be extracted. Thus, parameterizing the evolution of individuals' walking patterns into a common subspace allows for a more holistic, less biased, and straightforward analysis of primarily their overall differences in gait dynamics, inter- and intra-limb coordination over any differences attributed to absolute gait speed. Gait signatures can allow gait researchers to study or analyze the dynamical differences underlying impairment independently from gait speed, facilitating analysis of dynamics between individuals who may not be capable of walking at the same speeds and allowing investigation of changes in the underlying mechanism of gait changes under different conditions (walking speed, gait rehabilitation intervention, age etc.)

## Low-functioning stroke-survivors are less dynamically analogous to able-bodied and more dynamically variable compared to high-functioning stroke-survivors

Clinically, gait rehabilitation researchers use gait speed as a primary quantitative indicator of gait dysfunction [19,72,73]. While this coarse metric gives an overall value or number to one's overall gait function, it does not identify the specific impairments underlying the individuals' gait. To derive more precise measures or indicators of gait impairment, we anticipated that utilizing this gait signatures framework, we would be able to capture both subtle and obvious differences in kinematic patterns underlying impaired gait. In the clinic, stroke survivors are typically segmented into subgroups according to their self-selected walking speeds: high-functioning stroke survivors with a self-selected (SS) walking speed above 0.4m/s and low-functioning stroke survivors who adopt SS walking speeds less than 0.4m/s [74]. It is assumed that low-functioning stroke survivors are more impaired and thus adopt slower walking speeds to be able to navigate the environment safely. However, gait deficits of stroke survivors within either sub-group are heterogeneous across individuals and include different impairments such as foot drop, reduced paretic push-off during late stance, limited initial heel contact during early stance, as well as compensatory gait strategies such as hip circumduction and hip hiking. We expected that higher functioning individuals would have less severe impairments and would be more dynamically analogous to able-bodied individuals, whereas low-functioning stroke survivors would exhibit highly variable impairments from each other and be even less dynamically analogous to able-bodied dynamics compared to higher functioning stroke survivors.

To better visualize all developed individuals' gait signatures across their 6 different speed trials in our dataset, we again used MDS to project the 6D gait signatures to 3D. This mapping allows us to visualize the relative locations of individuals in comparison to all the other gait signatures to gain insights on how dynamically similar they are from one another. A 3D MDS gait map of all gait signatures reveals that able-bodied and high-functioning stroke survivors are located near each other, whereas low-functioning stroke survivors are farther and more dispersed and form distinct clusters in different regions of the map (Fig 3A). Sub-group level analysis reveals significant differences in the Euclidean distance metric (distance between each gait signature and the able-bodied centroid) between the able-bodied group and the low- and high-functioning stroke survivor groups, respectively (Fig 3B). Able-bodied gait signatures are located closest to the centroid, followed by high-functioning and low-functioning stroke survivors (Fig 3B). The within-group dispersion of gait dynamics for the low- and high- functioning

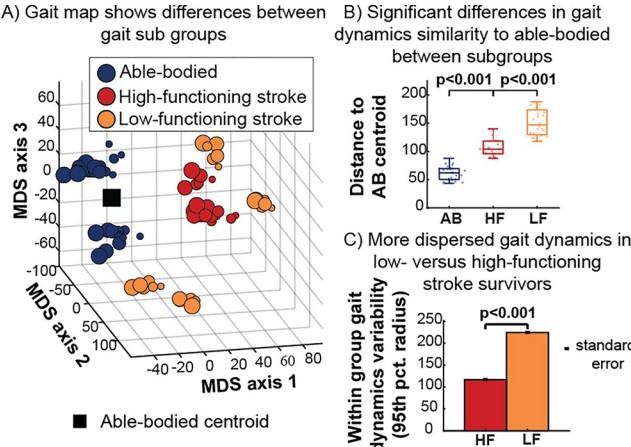

**Fig 3. Comparison of gait signatures across three gait subgroups: able-bodied (AB), high functioning (HF) and low functioning (LF).** A) 3D gait map using multidimensional scaling highlights the relative distances between AB (blue), HF (red) and LF (orange) stroke survivors. LF stroke survivors are less clustered and occupy distinct regions of the map away from the able-bodied centroid (black square). B) Gait dynamics similarity based on Euclidean distance between AB centroid and each participant, showing larger distances within the low-versus high-functioning groups. C) Within-group dispersion of gait signatures based on the radius of a hypersphere enclosing 95% of each group's gait signature reveals more dispersed gait signatures in low- versus high-functioning stroke survivors, highlighting the potential of gait signatures to capture individual differences in more severe gait impairments.

stroke survivors was calculated based on the radius of a hypersphere enclosing 95% of the groups' gait signatures. Using a leave-one-out sample with replacement method, multiple within-group dispersion calculations were conducted for each group and the average within-group dispersion was expressed alongside the standard error in Fig 3C. The 95th percent radius was significantly higher in the low-functioning stroke-survivors gait signatures compared to the high-functioning, highlighting that low-functioning gait signatures were more dispersed from each other (higher inter-individual variability) and the RNN model can capture these individual-specific gait deficits in individuals with more severe gait impairment.

## Gait signatures are biomechanically interpretable

While Principal Component trajectories and low-dimensional maps provide one way to compare the overall dynamics between individuals and groups, it remains to be seen what information the independent components of the 6D gait signature represent biomechanically. The contributions of each principal component (PC) to a gait signature fluctuates over the gait cycle, shown for an exemplar able-bodied, one high-functioning stroke survivor, and one low-functioning stroke survivor in Fig 4A. Superimposed individual stride-averaged PC projections from these 3 individuals (Fig 4B) highlight the specific differences in each PC. For PC1, both able-bodied and high-functioning stroke survivor traces are within the able-bodied 95% confidence interval, whereas the low-functioning stroke survivor is outside of these bounds around the middle of the gait cycle. For PC2, some regions of the low and high-functioning stroke survivor can be found outside of the confidence interval, however the entirety of the PC3 projection of the low-functioning stroke survivor is found outside of interval (vertically shifted). Given the generative nature of our RNN-based model, a specified number of the loadings on the PCs can be driven through the trained RNN model to reconstruct the corresponding kinematics. Thus, to interpret the individual PC components, the internal parameters corresponding to each isolated PC were driven through the gait dynamics model, generating gait predictions, i.e., a multi-joint coordination pattern and their temporal evolution over the

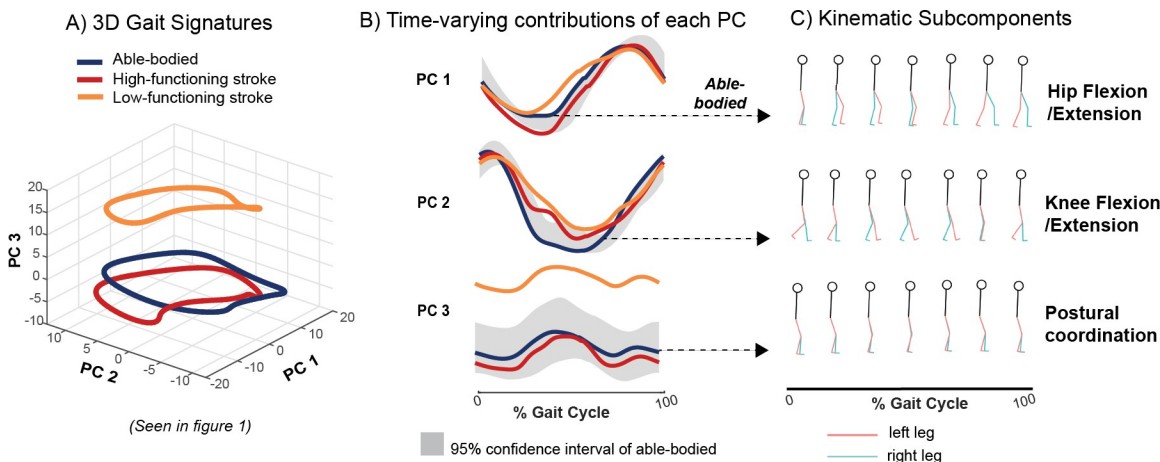

**Fig 4. Biomechanical interpretation of gait signatures.** A) Gait signatures reveal different gait dynamics between exemplar AB, low- and high-functioning stroke survivors. B) The loadings on each principal component (PC), e.g., the contributions of each PC vary over the gait cycle and can be compared to the AB 95% confidence interval (gray). C) Each PC generates specific multi-joint gait coordination patterns when used to drive the gait model, enabling biomechanical interpretation of gait deficits and effects of treatment.

gait cycle that can be visualized in an animation or gait movie. Stick figure snapshots (7 equally spaced samples of 100 frames) show that PC1 encodes dynamics driving hip flexion and extension, PC2 encodes dynamics driving knee flexion and extension and PC3 encodes dynamics driving primarily postural coordination (trunk location relative to joints) (S1–S4 Videos). This framework can potentially allow for the identification and targeting of individual-specific gait deficits, informing the tailoring of precision rehabilitation strategies.

## The gait dynamics model generalizes to unmeasured speeds

Our gait signature model can capture and predict nonlinear changes in dynamics in response to speed in cases where interpolation of kinematics may fail. We trained a different gait dynamics model using only 15s of data of the 2 fastest and 2 slowest walking speeds of each subject. Weighted averages of gait signatures from an individual walking at these four different gait speeds can be used to generate multi-joint kinematic trajectories that predict data from a gait speed that was not used to train the model (Fig 5). Predicted kinematics from interpolation of gait signatures across the four speeds resemble the measured kinematic reference more accurately than do the kinematics generated from interpolating gait kinematics, shown for an exemplary AB individual (Fig 5A) and low-functioning stroke survivor (Fig 5B). Kinematic prediction from interpolation of dynamics did considerably better than interpolating kinematics directly for the exemplary low-functioning stroke survivor shown in Fig 5B, indicating that interpolating gait signatures capture nonlinear (non-monotonic) changes in kinematics between speeds. The kinematic output of the interpolated kinematics follows that of the fast speed in the paretic hip closely but does not resemble the measured kinematic reference waveforms for the paretic knee or ankle angles. In some cases where interpolation of kinematics fails, the averaged dynamics do a better job at predicting kinematic trajectories at unseen speeds. Group level analyses show that the $R^2$ values between the measured and predicted kinematics from interpolated gait dynamics are significantly higher (Wilcoxon paired signed rank test) than interpolating kinematics within the able-bodied cohort (Fig 5C), but not for stroke (Fig 5D). In general, averaging gait dynamics produced less variable $R^2$ values and less $R^2$ outliers than averaging kinematics in both the able-bodied (Fig 5C) and stroke survivors (Fig 5D). The range of $R^2$ values in the able-bodied cohort for averaged dynamics was -0.20 to

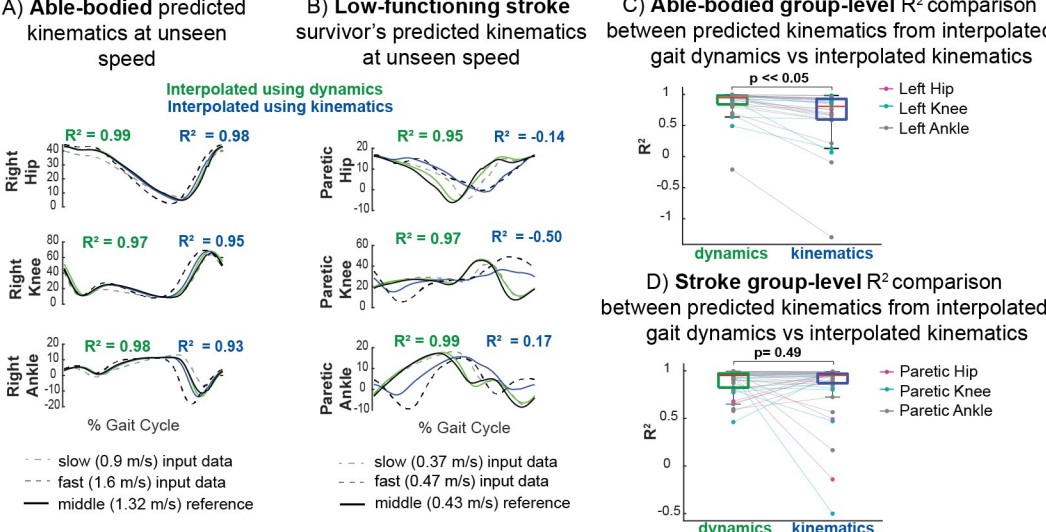

**Fig 5. Data-driven gait dynamics model predicts non-linear changes in joint kinematics with gait speed.** Gait predictions of joint kinematics (green) at intermediate gait speeds not used in model training were generated by interpolating gait signatures between slow (dashed grey) and fast speeds (dashed black) lines and using them to drive the gait model. Interpolated kinematics from gait dynamics (green) and interpolated directly from kinematics (blue) were compared to the measured reference kinematics (black solid). A) Predictions in an exemplar AB participant are more accurate when interpolating gait signatures compared to interpolating gait kinematics across speeds. B) In an exemplar low-functioning stroke survivor, interpolated gait signatures predict nonlinear changes in kinematics better at intermediate speeds than interpolated gait kinematics. Averaging the kinematics fail in this case where there are larger differences between the slow and fast speed paretic kinematics; the averaged kinematics (blue) follow the fast speed paretic hip kinematics whereas the other angles do not reflect waveforms that resemble either the fast or slow speed. The gait model can therefore predict movement reasonably well when interpolating between tested speeds. There is a statistically significant difference between group level $R^2$ comparisons (kinematics generated from interpolated dynamics vs interpolated kinematics) in the able-bodied (C) but not in stroke (D) cohorts. However, the range of $R^2$ values are larger in both able-bodied and stroke kinematic predictions resulting from interpolated kinematics (-1.30–0.98, -0.50–1.00 respectively) vs. predicted from interpolated gait dynamics (-0.20–1.00, 0.46–1.00 respectively). Thus, while the $R^2$ values may not improve on average for the stroke survivors, the model's performance is more robust overall.

1.00 compared to –1.30 to 0.98 in averaged kinematics whereas the range of R2 values in the stroke cohort for averaged dynamics was 0.46 to 1.00 compared to -0.50 to 1.00 in averaged kinematics. Two low-functioning stroke survivors show higher $R^2$ values of their hip, knee and ankle kinematic traces when interpolating kinematics vs. dynamics. Post hoc analysis revealed that these two stroke survivors (ST4 and ST2) were furthest away from the able-bodied centroid (least dynamically similar to able-bodied) as shown in Fig 2Bi. These results suggest that the RNN largely captures more stereotyped able-bodied dynamics and has a harder time learning the dynamics from more variable stroke individuals, especially those that deviate furthest from able-bodied. We acknowledge that our model likely is not capable of generalizing to speeds beyond the ranges of the input data (extrapolating), as RNNs are highly dependent on the training data that it sees to learn patterns in the dataset. One benefit of this capability, however, is that any data that deviates from the walking speeds in the training set can still be analyzed, reducing the number of speeds required in the training set to achieve a model that is valid across a range of speeds. Additionally, our small sample size limits the amount of data the RNN sees for each diverse type of stroke dynamics, thus, with a larger sample size of stroke survivors and longer trials, the RNN may be able to make better kinematic predictions of lower-functioning stroke survivors. Moreover, this result highlights the utility in predicting kinematics in unseen conditions which in contrast cannot be made using discrete biomechanical or clinical metrics, nor with current biophysical models.

## Gait sculpting: Manipulating the PC components of an individual's gait signature identifies specific coordination deficits in stroke survivors

Previously, we showed that we can leverage our model to reconstruct the kinematics of healthy PC projections of the gait signature to gain insight into their independent biomechanical interpretations. However, identifying and interpreting the biomechanics related to impaired PC dynamics of stroke-survivors' gait would prove to be even more beneficial, as these dynamics can potentially serve as rehabilitation targets when designing tailored gait intervention/strategies for individuals. Here we present an example of how we use gait signatures to identify specific biomechanical or coordination targets in specific stroke survivors. Specifically, we utilize our finding that the phase-varying contributions of the 6 principal projections of the gait signature differ in individual-specific manners (Fig 6A). For example, AB2's 6 PC contributions all lie within the 95% confidence interval of all able-bodied individuals. ST4 primarily shows

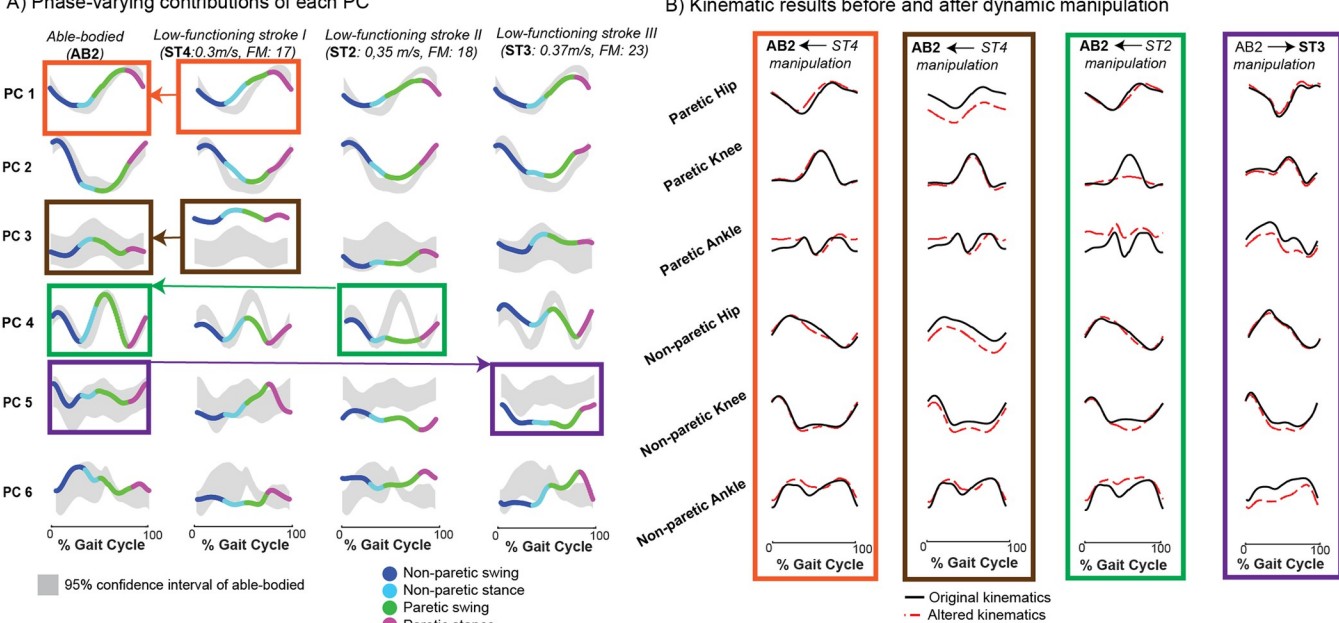

**Fig 6. Gait sculpting: interpolating between components of able-bodied and stroke gait dynamics to visualize anticipated gait improvement.** The components of individuals' gait signatures can be manipulated (gait sculpting) to understand the relationship between specific underlying dynamics and their corresponding kinematic phenotype. A) The projection on each of the 1st 6 principal components (PCs) can be observed for a representative able-bodied (AB2), two low functioning stroke-survivors each having similar self-selected (SS) speeds and Fugl-Meyer (FM) scores (ST2 & ST4, as denoted in Fig 2) and another low functioning stroke survivor (ST3) who has a higher FM score and faster SS walking speed. The PC projections are colored according to the 4 gait phases (non-paretic swing, non-paretic stance, paretic swing, paretic stance). The right leg of the unimpaired individuals was arbitrarily assigned to be paretic and the left leg, non-paretic for consistency. Colored boxes and arrows (orange, brown, green, purple) show specific, single PC manipulations, for example, the orange boxes and arrow illustrate that the PC 1 projection of AB2 was replaced with the impaired PC 1 projection from ST4. B) The AB2:ST4 manipulation (orange) shows how AB2's original phase averaged kinematics (black) was manipulated by ST4's impaired PC 1 projection (red dashed traced). ST4's impaired PC 1 manifests in AB2's healthy kinematics showing deviation primarily in the hip kinematics (as suggested in Fig 4 where healthy PC 1 encodes a kinematic subcomponent corresponding to hip flexion/extension) and some deviation in the ankle angles, especially the paretic ankle. The AB2:ST4 manipulation (brown) shows how ST4's impaired PC3 manifests in AB2's healthy kinematics; we observe a vertical shift downwards (red trace) of the bilateral hip angles as well as the non-paretic knee. This change in hip flexion highlights that this impaired PC3 encodes a reduction in the hip flexion angles; pointing to a more crouched gait (trunk is leaning forward more). The AB2:ST2 manipulation (green) shows replacing AB2's PC4 projection with ST2's impaired PC4 dynamics shows deviation in the knee joints especially during paretic swing, a vertical shift upwards in the paretic ankle angle kinematics and deviations around the middle of the gait cycle (transition between non-paretic stance and paretic swing) in the non-paretic ankle kinematics. Alternatively, the AB2:ST3 manipulation (purple) the impaired PC5 in ST3 is replaced with the healthy PC5 projection from AB2 resulting in slight increase in non-paretic knee magnitude and reduced amplitude of paretic and non-paretic ankle flexion. The result of this manipulation points to potential predicted improvements (or deviations) that can occur when aiming to mimic PC5 healthy dynamics in this stroke survivor allowing offline in-silico testing of potential avenues for gait rehabilitation for this stroke survivor.

major deviation from AB in PC 3 (located entirely above the AB confidence interval), impaired dynamics during paretic swing in PC 4 and overall irregular shapes in PC 5 and 6. ST2's PC3 is largely within the AB confidence interval, however PC 4's paretic swing shows major deviation, their PC 5 contribution is shifted below the AB confidence interval and PC 6 shows an irregular shape. ST3's PC5 projection lies below the AB confidence interval and PC 6 projection is irregularly shaped compared to AB. To validate our finding that suggested that PC 3 primarily influences hip flexion or extension, we exchanged AB2's healthy PC1 with that of ST4 (Fig 6A, orange boxes and arrow) and we observed if and how AB2's original hip joint kinematics (Fig 6B, orange box, black trace) deviated (Fig 6B, orange box, red dashed trace and S5 Video). To gain further insight into how the other PC deviations manifest in movement, we manipulated the PC3 projection of AB2 by replacing it with that of ST4 (Fig 6A, brown boxes and arrow). The kinematic reconstruction from this manipulation (Fig 6B, brown box, red dashed trace and S6 Video) shows a vertical shift downwards for bilateral hip angles and the non-paretic knee. The vertical shifts in the hip flexion/extension angles suggest a major difference in this individual's posture (perhaps stroke individual leaned forward more during gait) compared to able-bodied. We manipulated AB2's PC4 projection by replacing it with that of ST2 (Fig 6A, green boxes and arrow). This manipulation affected specifically the paretic and non-paretic ankle angles and both knee joints primarily during the period between non-paretic stance and paretic swing (Fig 6B, green box, red dashed trace and S7 Video). This result highlights a coordination deficit between these specific joint angles and, if targeted accurately, may allow for corrected gait patterns of this stroke survivor. Conversely, we tested the effects of replacing an impaired PC projection with a healthy one to observe how gait impairments can potentially be improved. We replaced ST3's PC5 with that of AB2 (Fig 6A, purple boxes and arrow) and observed a substantial change in the magnitude and shape of bilateral ankle angle trajectories and slight increase in non-paretic knee magnitude (Fig 6B, purple box, red dashed trace and S8 Video). We can infer that to make improvements to ST3's PC5 towards able-bodied or normative kinematics, rehabilitation focusing on these specific knee and ankle strategies may prove useful.

### Self-driven signatures: Our gait dynamics model revealed robustness of gait predictions establishing the utility of gait signatures in precision medicine

The ability to predict future kinematics based on measured data is key to rapid, virtual design of personalized interventions. We demonstrate that the recurrent neural network model of gait dynamics, once primed with several gait cycles of data from either able-bodied or stroke participants, can predict future joint angle trajectories (Fig 7). Once the network is primed, an initial posture is presented (initial condition, denoted by blue vertical bar) after which the model self-drives i.e., predicts the general shape of future kinematics in a feedforward manner (without referencing previous measured data points) in an able-bodied (Fig 7Ai, left) and stroke individual (Fig 7Bii, left). A smooth transition is seen between the previously measured gait cycle (green) and the self-driven cycle (red trace) for both AB and stroke (Fig 7Ai, right and 7Aii, right respectively).To verify that the model was not generating a gait cycle prediction entirely by chance, we calculated the Euclidean distance between the kinematics of the predicted (self-driven) gait cycle and the kinematics from each of the measured gait cycles. We computed the distribution of distances between each predicted gait cycle with all other gait cycles from the same individual (Fig 7B, purple bars). We then compared the predicted gait cycle to the target gait cycle (Fig 7B, red bars). In the able-bodied individual, the predicted gait cycles are more similar to the target gait cycle (Fig 7Bi, red bar) than 79% of all gait cycles. However, in the stroke survivor, 60% of other gait cycles were more similar to the predicted

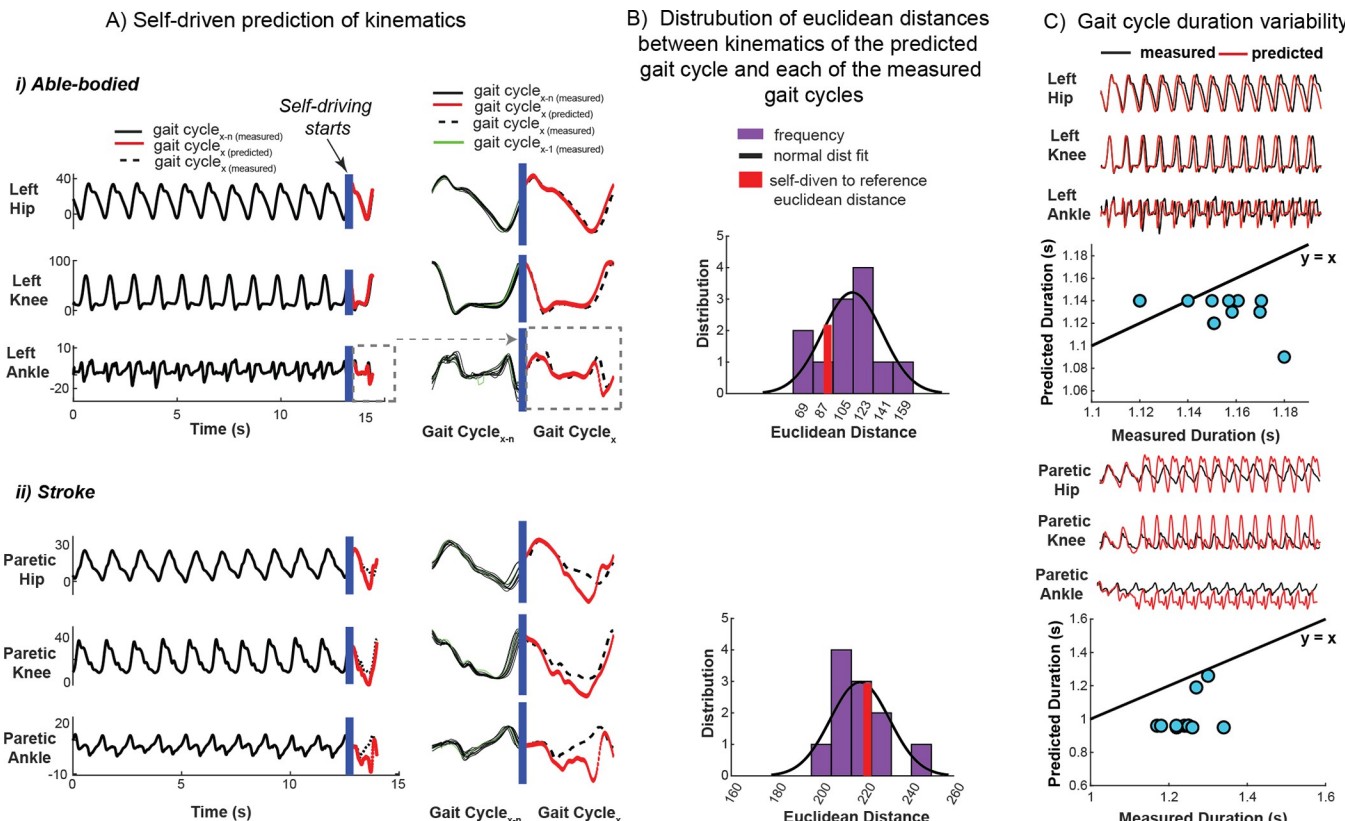

**Fig 7. Our trained RNN model can predict the time evolution of kinematics from an initial posture.** The trained gait dynamics model can predict individual-specific time-evolution of gait kinematics from an arbitrary initial posture (self-driving) in able-bodied (A, i) and stroke (A, ii) once the network is primed with several gait cycles of an individual's data (gait cycle$_{x-n(measured)}$, black solid). This predictive ability shows that the model encodes the gait dynamics underlying movement. Despite inter-cycle kinematic variability, the gait dynamics model can predict the general shape of the next gait cycle of kinematics (gait cycle$_{x(predicted)}$, red) in an able-bodied individual (A, i) and stroke survivor (A, ii), however, predicted kinematics (red) show larger deviation from the measured reference gait cycle (gait cycle$_{x(measured)}$, black dashed) in the stroke survivor. A smooth transition exists between the measured kinematics from the gait cycle preceding (gait cycle$_{x-1(measured)}$, green) the self-driven predicted cycle (red). For the representative able-bodied individual (B, i), the Euclidean distance (deviation) between the predicted gait cycle of kinematics and its respective measured kinematics (reference) is ~79% lower than the distance between the other gait cycles in the trial; ruling out that the kinematic predictions are attributed to chance. The deviation (Euclidean distance) of the predicted gait cycle of stroke (B, ii) kinematics to its reference gait cycle is ~40% lower than the distance between the other gait cycles in the trial. This suggests that the dynamical model is less able to accurately predict stroke kinematics better than chance. The dynamical model was first initialized with all the trial's kinematics data (15 seconds) (black trace) after which the trial's initial posture was presented to the model to self-drive kinematics (red trace) in feedforward mode for 15 seconds (C, i, top plot). The duration of each gait cycle from the measured kinematics is not well encoded by the dynamical model; gait cycle durations of the predicted kinematics are typically underestimated in both able-bodied (C, i, bottom plot) and stroke (C, ii, bottom plot) (to a larger degree) in self-driving mode and as such deviate from the y = x reference line (black).

gait cycle than the target gait cycle (Fig 7Bii, red bar). This suggests that the model is less able to accurately predict future kinematics in stroke gait. Note, to calculate the Euclidean distances between the gait cycles, we need to normalize the period of each gait cycle to the period of the self-driven cycle. To avoid the potential of bias due to this normalization, we also performed a comparison using a metric that was not manipulated in time–gait cycle duration. After priming the model, we presented the model with the first posture of the trial and ran the network forward in self-driving mode for the remainder of the trial length (15 seconds). Able-bodied self-driven predicted kinematics resembled the reference kinematics closely (Fig 7Ci, top plot) whereas stroke self-driven predicted kinematics matched the first gait cycle closely but soon converged to patterns reflecting able-bodied kinematics (Fig 7Cii, top plot). The gait cycle duration of the first few cycles of the self-driven kinematics match those of the measured kinematics (blue dots located close to the y = x line) in both the exemplary able-bodied (Fig 7Ci,

bottom plot) and stroke individual shown (Fig 7Cii, bottom plot), however kinematics soon diverged to shorter and relatively consistent gait cycle durations (blue dots appearing almost horizontal in the plots) result in both cases. The model may preferentially predict able-bodied kinematics, which were less variable between individuals than were post-stroke kinematics. These results highlight that the model encodes gait dynamics that can predict kinematics over short timescales, but the variability and amount of training data may influence predictive power over long timescales. Our model provides a foundation for a sample-specific gait dynamics model to predict the effects of environmental perturbations, assistive devices, and treatments without extensive experimental sessions. In-silico predictions using the gait dynamics model may thus reduce the experimental time, cost, and participant burden required for personalized gait characterization, treatment personalization, and device design.

## Discussion

### Summary

Here we establish a data-driven framework for comparing and predicting individual-specific loco-motor patterns without needing to construct physiologically based mechanistic models. As an initial proof of concept, complex neuromechanical gait dynamics were modeled using a relatively simple recurrent neural network that captures the rules by which joint kinematics during gait transition from one time point to the next. Because the network was trained on multiple healthy and impaired individuals walking at several speeds, its internal parameters provide a basis for comparing, interpreting, and predicting gait dynamics. Gait signatures further capture coordination between joints and limbs without the need for pre-selecting gait features that may introduce bias and ignore the continuous nature of gait. We show that individuals have little variance in gait dynamics across speeds, leading to the individual-specific "gait signature" concept and enabling comparisons between individuals moving at different speeds. Across stroke survivors, we found greater heterogeneity in low-functioning individuals who exhibited disparate gait dynamics despite similar clinical metrics, highlighting the potential utility of gait signatures in providing more sensitive diagnoses to personalize therapies. Gait signatures provide a predictive simulation framework for sculpting gait dynamics to understand coordination deficits and predict kinematics, potentially forecasting the effects of rehabilitative devices or treatments. Finally, the gait signatures methodology can be readily applied to other periodic motions across species and across conditions that alter movement and may be a powerful adjunct to modern experimental methods aimed at understanding the neural mechanisms underlying movement.

### Computational framework captures the neuromechanical dynamics of walking

Using a data driven modeling approach enabled us to learn the underlying gait dynamics based on data rather than constructing a neuromechanical gait model based on first principles. Data-driven approaches in gait have not focused on gait dynamics but have solved tasks based on unique features in multi-dimensional gait data such as classifying gait based on pathologies [75] or conditions such as fatigue and non-fatigue [76], identifying gait events (e.g., initial contact, loading response [77–79]), and discriminating individuals [62,80]. Gait dynamics have typically been described though neuromusculoskeletal models based on physical principles focusing on musculoskeletal mechanics, [30,81] but they lack adequate representations of the neural systems that contribute to the resulting movement patterns, particularly in neurological impairments such as stroke [31]. Machine learning methods to capture dynamics have been used across physics, engineering, and neuroscience to learn the dynamics underlying complex

systems when the governing equations are unknown [58,82,83]. Recently, machine learning models have been used in human gait to predict continuous kinetic variables such as ground-reaction forces [84] or joint torque [85,86] based on kinematic data. Dynamical machine learning models have also been used to encode gait dynamics, including responses to perturbations or assistive devices, but their model structure did not enable comparisons between individuals [87–91]. Here, our RNN-based gait dynamics model provides a means to capture the rules underlying continuous, multi-joint coordination between bilateral lower limb joints, and how they evolve over time. Accordingly, we do not explicitly capture mechanical dynamics (i.e., the relationship motion and force), but the effects of force interactions within the body and environment and implicitly represented in how multi-joint kinematics evolve over time, with the network parameters and the internal states at each time point determining the output kinematics.

As gait arises from complex interactions between the nervous system and the musculoskeletal system that are not easily modeled from first principles, a data-driven approach provides a powerful framework for capturing and comparing neuromechanical constraints on gait dynamics. While biomechanical dynamics clearly play a role in movement, the activation of muscles by the nervous system enables the body to perform a variety of motor behaviors. However, the governing spatiotemporal dynamics of neuromuscular signals are poorly understood, especially in neuro-pathologies such as stroke. During behaviors such as locomotion, motor patterns can be characterized by the number and structure of motor modules, or muscle synergies, defining groups of co-activated muscles producing a biomechanical function for gait [92]. Similar motor modules are used within individuals across different task conditions [93–95], and are shaped by learning and disease [96,97]. Particularly in post-stroke gait, motor modules appear to constrain motor function. Fewer motor modules are observed post-stroke with the number of modules correlated to reduced walking speed [98,99]. Further, different patterns of motor module merging are seen in slower walking stroke survivors, differentially affecting gait biomechanics in a manner that may necessitate individualized rehabilitation approaches [100]. Adding neural constraints such as motor modules on muscle activations in musculoskeletal simulations improve predictions of key physiological variables such as joint loading in osteoarthritis [101]. However, relating motor modules to kinematic gait patterns post-stroke and in other neurological disorders has been challenging, likely because the neural constraints are underspecified [27,102–105]. Corroborating results from motor module analysis, there were greater differences in gait dynamics amongst the slowest walking stroke survivors. Since the gait signatures capture spatiotemporal constraints underlying gait dynamics, they provide a complementary approach to musculoskeletal simulations. Ultimately, gait signatures may play a complementary role to biophysical simulations, enabling the relationships between biomechanical principles, neural constraints, and the emergent gait dynamics to be revealed.

## Gait signatures enable holistic comparison of gait dynamics across individuals, speeds, and groups

In contrast to other applications of dynamical machine learning models for gait, we capture multiple individuals within a single network, enabling comparisons of gait dynamics across groups, individuals, and gait conditions. Rather than using the network as a black box solely to generate predictions, we explicitly compare and interpret the model's internal parameters to identify low-dimensional latent variables representing gait dynamics. To encourage a generalizable data-driven gait dynamics model, we omitted subject and trial condition (gait speed) labels as inputs to the neural network. Adding input labels might force the RNN to create separable gait models, whereas our goal was to have the network learn a structure that could be modified parametrically to represent individual differences in the neuromechanics of walking.

Similarly, neuromusculoskeletal models assume common dynamic principles across individuals, using parameter variations to represent individual differences [27,28,103,104,106]. We intentionally designed a relatively simple RNN architecture (e.g., single layer, linear input/output) as a starting point to recover as much interpretability as possible, with the awareness that more complexity could be added to the model architecture (number of hidden layers, number of neurons, etc.) if required to fit a given data set robustly. The representations of gait dynamics that emerge from our model holistically capture the changes underlying measured kinematics, without being attributable to specific neural or biomechanical constraints. The loss of physiological interpretability is counterbalanced by the holistic approach to representing gait dynamics and explaining gait kinematics features.

Analogous to written signatures, we find that features of individual-specific gait signatures are largely preserved across walking speeds. Recognizable qualitative features of handwriting are preserved even as the size of letters changes quantitatively, or if different limbs, or writing instruments are used [107]. Similarly, it is well known that individuals can be recognized based on how they move or walk [28,40,108–110], even if joint angle excursions are similar. We show that gait dynamics are more similar within individuals across speeds than between individuals, leading to the concept of the gait signature. In contrast, gait kinetics and kinematics vary characteristically across speeds, such that they cannot be directly compared across speeds [111]. The relatively small changes in gait signatures across speeds suggest that the signatures reflect changes in the spatiotemporal relationships between joint kinematics, rather than quantitative changes in their magnitude. As such, gait signatures appear to encode individual-specific constraints of walking, making it possible to compare gait either within or between individuals walking at different speeds.

Gait signatures characterize the high inter-individual variability in gait impairment amongst stroke survivors beyond overall gait function explained by clinical gait metrics. This heterogeneity is a direct reflection of the wide range of impairments in stroke survivors, including muscle weakness, impaired coordination, spasticity, abnormal synergistic activation (muscles not independently coordinated), and compensatory motion [19,53,112]. We found that higher-functioning stroke survivors were more dynamically similar to each other, whereas lower functioning stroke survivors were more dispersed. In fact, two low-functioning stroke survivors with similar clinical metrics (Fugl-Meyer score and gait speed) had quite different gait signatures. As such, gait signatures have the potential to provide insights into individual differences in gait dynamics that are simply not captured by clinical metric such as gait speed. Moreover, in contrast to higher-functioning stroke survivors who share similar gait dynamics, lower-functioning stroke survivors may require more individuals individualized rehabilitation approaches targeting specific aspects of gait dysfunction. Further gait signatures do not require a priori selection of which gait variables to compare [113–116]. As such gait signatures provide a powerful, holistic approach to enhance the specificity and precision of gait diagnosis and treatment. Our study inclusion criteria exclude severe contractures or deformities that interfere with normal ambulation and in future work the gait signatures would need to be interpreted and correlated with clinical evaluation of strength, range of motion, sensorimotor impairment, and/or limb deformities. The demographic and clinical information of the stroke participants in our study are available in Supplementary Materials S1 Table. Gait signatures could be part of a set of multi-modal data to account for the diverse causal factors underlying each individual's gait pattern (e.g., lesion neuroanatomy, medical confounding variables, musculoskeletal conditions, psychosocial variables, physiological contributors to gait and environmental factors). This framework can potentially extend to other diseases, disorders, injury, etc. to gain further insight into individuals' specific impairments and uncover specific targets towards developing targeted therapies for individuals.

## Gait signatures enable biomechanical interpretation and manipulation

Our gait dynamics model enables biomechanical interpretation of gait signatures and exploring "what if" scenarios to sculpt desirable gait dynamics. Gait signatures are based on principal components (PCs) of the gait model internal states, where the weightings on each PC vary over the gait cycle. The model parameters can be prescribed over the gait cycle, resulting in the predicted kinematic outputs (i.e., joint angles). The gait signature PCs and their time-varying weightings can be individually prescribed in the network as a method to reveal the specific inter- and intra-limb coordination patterns governed by each PC. Further, any combination of PC's can be combined and reweighted to generate new kinematic output patterns. For example, we interchanged healthy and impaired PCs to gain deeper insight into how specific impaired PCs alter healthy gait and vice versa. Further, interpolating gait dynamics can predict gait kinematics at walking speeds that were not used in the training data. Especially when there was a nonlinear response in gait kinematics across speeds, interpolation of gait dynamics to predict gait kinematics performed better than interpolating gait kinematics directly. As such our data driven gait dynamics model can be used to show how changing select components of the gait signature alters gait kinematics, providing a potential framework to identify personalized therapeutic targets for gait rehabilitation.

## Gait signatures have potential to predict future kinematics

Another powerful aspect of our gait signatures framework is its ability to generate future gait kinematics in the absence of new data. The model is self-driving for able-bodied individuals, predicting multiple cycles of gait kinematics in the future. However, the ability of the model to predict future stroke kinematics is limited to approximately one gait cycle in the future; rendering it promising in applications that provide control signals to rehabilitation devices (e.g., exoskeletons). Furthermore, our model is likely only capable of generalizing to speeds within the speed ranges of the input data. There was a moderate association between participants' similarity to able-bodied gait signatures (distance to the able-bodied centroid) and the RNN's ability to predict gait kinematics over one gait cycle (S6 Fig). This association is likely due to the RNN favoring able-bodied dynamics during model fitting, which were more homogeneous than those of high- or low-functioning stroke survivors. A larger post-stroke sample may improve the RNN's ability to encode and predict pathological gait dynamics. Further, the reduced predictive power for the stroke participants can be attributed to our model architecture's relative simplicity and short time-series (15 seconds/ 1500 sample points per trial). These factors should be addressed to improve the predictive capacity of the model for impaired gait in the future. Additionally, including more variables besides sagittal plane kinematics (e.g., frontal plane and coronal plane kinematics and joint forces, may improve learning of the underlying dynamics of gait and increase predictive capability of our model.

## Generalization to other species and rhythmic movements

Because the input to this model are periodic sequences of behaviors, our gait dynamics framework should generalize to other species that display similar behavioral motions (e.g., flight, crawling, and walking). Physicists, computational biologists, and other scientists can benefit from this method by studying the dynamical behavior of species whose neuromechanical models and physics of complex terrains are difficult to model. This is the first study to our knowledge that uses a neural network to study the dynamics of gait in an interpretable manner. While much work is left to be done, we have provided a simplistic, unsupervised framework to discover individual-specific differences in walking in health and disease in humans. Despite being limited by a small dataset, we have shown that our model is generalizable to

characterizing and predicting kinematics of one held-out subject using leave one out cross validation (S3 Fig). Here we focus on demonstrating the innovation, feasibility, and potential advantages of our RNN gait signature approach, justifying the need and potential for further development by scaling to larger-sample studies.

Importantly, this methodology relies on having a periodic or quasi-periodic pattern, as non-periodic patterns would not be able to generate a phase and subsequent signature. We also limited our inputs to gait kinematics, anticipating applications to the proliferation of new measurement modalities for movement in humans and animals such as wearable sensors and markerless video-based motion capture [117–119]. However, the gait signatures framework could easily be extended to include other data types (e.g., force, muscle activity, joint loadings, center of mass dynamics) and experimental conditions (overground walking, biomechanical constraints, gait interventions, such as exoskeletons, functional electrical stimulation, or treatment e.g., drugs, optogenetics). In practice, a more comprehensive data set would be needed within each gait group to train a model capable of capturing the full range of variability in gait dynamics. Short of having a massive data set, it may also be possible to leverage synthetic gait data from simulations to span the full range of feasible gait dynamics variations.

Overall, by modeling the dynamics of individual's gait based on measured data, we uncovered individual-specific representations of individuals' neuromechanical constraints that allows direct comparisons between individuals who do not walk at the same speed. The gait signatures framework has implications for the diagnosis of disease, development of future tailored gait therapies or interventions and tracking meaningful changes in the fundamental neuromechanical mechanism of walking.

## Materials and methods

### Ethics statement

All participants provided written informed consent prior to study participation, and the study protocol was approved by the Emory University Institutional Review Board.

### Human subject participants

To develop dynamical signatures of human gait, we collected data in seven post-stroke individuals (age = 56 ± 12 years; 2 females; 48 ± 25 months post-stroke; Lower Extremity Fugl-Meyer = 20 ± 4) and five able-bodied (AB) controls (age = 24 ± 4 years; 4 female). All post-stroke participants experienced a cortical or subcortical ischemic stroke, were able to walk on a treadmill for one minute without an orthotic device, and exhibited no signs of hemi-neglect, orthopedic conditions limiting walking, or cerebellar dysfunction.

### Experimental design

Participants completed 15-second walking trials at six different speeds, distributed evenly between and ranging from each participant's self-selected (SS) speed to the fastest safe and comfortable speed. Across stroke participants, gait speeds ranged from 0.3–1.6 m/s. Each participant's fastest walking speed was determined by progressively increasing the treadmill speed from the SS speed until the participant could no longer comfortably or safely maintain the speed for 30 seconds. Participants rested for 1–2 minutes between consecutive gait trials. During data collection, speed increased from the participant's SS to their fastest speed (i.e., not randomized).

## Data acquisition

Reflective markers were attached to the trunk, pelvis, and bilateral shank, thigh, and foot segments [120]. We collected marker position data while participants walked on a split-belt instrumented treadmill (Bertec Corp., Ohio, USA) using a -7-camera motion analysis system (Vicon Motion Systems, Ltd., UK). Participants held onto a front handrail and wore an overhead safety harness that did not support body weight. Marker data were collected at 100 Hz, and synchronous ground reaction forces were recorded at 2000 Hz and were down sampled to 100Hz using previously established techniques [121–123].

## Data processing

Raw marker position data were labeled, gap-filled, and low-pass filtered in Vicon Nexus. Labeled marker trajectories and ground reaction force raw analog data were low-pass filtered In Visual 3D. Gait events (bilateral heel contact and toe-off) were determined using a 20-N vertical GRF cutoff, and sagittal-plane hip, knee, and ankle joint kinematics were calculated in Visual 3D (C-Motion Inc., Maryland, USA).

## RNN model development

Our goal was to start with a simple RNN to reduce overfitting with too many parameters and deep layers. We wanted the simplest model capable of learning the dynamics underlying gait which also preserved interpretability. The simplest recurrent neural network (RNN) model architecture consisted of one input layer, one hidden layer and one output layer. The hidden layer was composed of long short-term memory (LSTM) units with a lookback parameter that spanned at least one gait cycle. Model hyperparameter selection is described in a later paragraph.

## RNN model training

Model fitting on our selected dataset and architecture was executed on the order of minutes to tens of minutes, using Keras 3.7.13 and TensorFlow 2.8.2 on Google Colab's standard GPU with high-RAM runtime (54.8 gigabytes). The RNN model was trained using bilateral, sagittal-plane, lower-limb joint angles from 5 able-bodied (AB) participants and 7 stroke survivors each walking on a treadmill at 6 steady speeds, ranging from each participant's preferred speed to the fastest safe speed. Our training dataset was input to the RNN in multivariate format (not concatenated) [62,63]. We trained a sequence-to-sequence RNN with 512 long-short-term memory (LSTM) activations units in the single hidden layer, capable of using 15 seconds (sample rate of 100 Hz) time-series kinematic input data (0 to T-1) to predict kinematics one time-step in the future (1 to T) for all training data across individuals and speeds. Our data was batched according to the number of total trials (N = 72); thus, the LSTM maintains its internal state while a batch is being processed, after which the internal state can be maintained or cleared. Because our network retains its internal state from one time step to the next (i.e., the RNN is *stateful*), we have fine-grained control over when the internal state of the LSTM network is reset. The input data from all trials was 'mini-batched' into 2 training batches and 1 validation batch (499 samples each) that were used to update model weights on each model run (epoch). To format our data into equal length input and output mini batches for training and account for the output data being a one-time step shifted version of our input data, our lookback parameter must be one value less than a divisor of the trial length. For example, in our dataset (1500 sample length trials), a lookback parameter of 499 would result in the first mini batch input of samples [0:499] which will predict our reference output samples [1:500], our 2nd mini batch input data would include samples [501:999] and corresponding output

[502:1000] and the last mini-batch input of samples [1001: 1499] predicts samples [1002: 1500]. This lookback parameter of 499 allows us to construct 3 mini-batches of shorter input and output data lengths which would be used to train and validate the RNN model (2:1 training: validation mini batch split). Similarly with the lookback parameter of 499 (2:1 training: validation mini-batch split) and 749 (1:1 training: validation split). Mean squared error was used as the LSTM loss function and ADAM as the optimization algorithm because it is fast, has a small memory footprint and is well suited for large-parameter deep learning models [124]. The model was trained for at least 5000 iterations or until training and validation error converged ($< 0.75$˚). The training resulted in a sample-specific dynamical model structure defined by a single set of LSTM network weights ($W$). The model's internal states capture trial-specific dynamics predicting the time evolution of joint kinematics; activation coefficients ($H$) and memory cell states ($C$) and are tuned based on kinematic input.

## Model hyperparameter selection

We selected the hyperparameter values of 512 nodes in the LSTM layer and a 499-sample LSTM lookback length (number of samples preceding the current time point that is used to train the LSTM) were selected based on training and validation loss, as well as the ability to encode dynamics over short and long timescales. In two steps, we evaluated all pairs of the following hyperparameter values: 1024, 512, 256, and 128 LSTM nodes and 749, 499, and 249-sample lookback parameters. Because RNN performance can change with the parameters used to initialize the RNN, we fit an RNN gait dynamics model 20 times using random initial parameters, for each hyperparameter pair. First, we compared model training and validation loss for each hyperparameter pair: the 'best' hyperparameter pair would have low training and validation loss. The following [node-lookback] pairs were considered the *best* hyperparameter pairs: 512–499 ($MSE_{train} = 0.010 \pm 0.001$ deg$^2$; $MSE_{val} = 0.018 \pm 0.000$ deg$^2$), 256–749 ($MSE_{train} = 0.010 \pm 0.002$ deg$^2$; $MSE_{val} = 0.015 \pm 0.001$ deg$^2$), 256–499 ($MSE_{train} = 0.010 \pm 0.001$ deg$^2$; $MSE_{val} = 0.017 \pm 0.000$ deg$^2$). (S1A Fig). The training loss was not different between hyperparameter pairs ($p > 0.235$). The validation loss differed between all three models ($p < 0.001$), with the 256–749 model having the lowest validation loss. However, if the differences in validation loss of less than 0.003 deg$^2$ corresponded to meaningful differences in performance was unclear.

Our second analysis was, therefore, used to compare the three hyperparameter pairs deemed *best* in the prior analysis. Here, we evaluated the models' abilities to encode the average dynamical behavior over long timescales (*long-time*) and the stride-to-stride behavior (*short-time*). We defined the *best* model as the one with the highest long- and short-time performance. The following analysis was performed for 10 of the 20 random initializations. For long- and short-time analyses, we created a single set of *reference* dynamics as done in the manuscript: we performed one time-step predictions over the full (1500-sample) time-series. This step provided best-case predictions of the gait dynamics (S1B Fig).

**Long-time performance.** We generated long-time predictions of each trial's gait signatures (RNN latent states) by simulating each participant's gait dynamics forward in time, 1500 samples into the future. Each simulation was initialized by setting the RNNs' latent states to those of the last sample of the trial's reference dynamics and using the last sample of the trial's kinematics. We then phase-averaged both the reference dynamics and the long-time predictions using the same technique as described in the main manuscript. Long-time performance was defined as the similarity of the phase-averaged latent states (*i.e.*, the gait signatures) between the reference and the long-time predictions and was quantified using $R^2$. Note that using $R^2$ as a similarity metric, rather than the Euclidean distance metric used in the main manuscript, was needed to compare models with different numbers of nodes. Unlike $R^2$,

Euclidean distances are sensitive to the number of samples used to compare models, which would bias short- and long-time performance towards models with fewer nodes. Low $R^2$ values between predictions indicates that the learned dynamics are sufficiently complex to capture instantaneous gait dynamics but can also accurately generate the time evolution of dynamics over the gait cycle—a major challenge in data-driven models of locomotion [89–90].

The 512-node model captured gait dynamics over long time scales significantly better (*i.e.*, more accurate predictions of the time-varying dynamics) than the 249-node models (S1B Fig). For long-time predictions, the 512-node model predictions ($R^2 = 0.50 \pm 0.46$) were better than the 249-node 499-sample lookback model ($\Delta R^2 = 0.27 \pm 0.06$; $p < 0.001$; independent-samples t-tests) and the 249-node 499-sample lookback model ($\Delta R^2 = 0.31 \pm 0.07$; $p < 0.001$).

**Short-time performance.** We generated short-time predictions by simulating single strides in each trial's time-series, initialized from the first sample of each stride. Initialization used the latent RNN states and kinematics of the reference dynamics at the onset of a new stride (phase = 0 rad). For each initial condition, we integrated the dynamics forward in time, up to the onset of the next stride. For each stride, we then compare the similarity of the reference dynamics to the dynamics of the corresponding short-time prediction using $R^2$. Short-time performance was quantified as the average $R^2$ value across trials for a single model and initialization.

The 512-node model captured gait dynamics over short time scales significantly better (*i.e.*, more accurate predictions of the time-varying dynamics) than the 249-node models (S1B Fig). For short-time predictions, the 512-node model predictions ($R^2 = 0.11 \pm 0.51$) were more accurate than the 249-node 499-sample lookback model ($\Delta R^2 = 0.51 \pm 0.13$; $p = 0.055$; independent-samples t-tests) and the 249-node 499-sample lookback model ($\Delta R^2 = 0.34 \pm 0.09$; $p < 0.001$). Based on difference in short- and long-time prediction performance, we selected the 512-node, 499-sample lookback hyperparameters for the RNN model.

## Leave-one-out subject model evaluation for generalizability

Using the selected hyperparameters, 12 different models were trained where one different subject (all 6 speed trials per subject) was held out for evaluation on each model run. The same model architecture, training and validation setup was used as the original model trained using the full dataset (12 subjects). The minimum training loss, validation loss, and overall evaluated test loss for each model were extracted and box plots of each generated. The Wilcoxon Rank-Sum Test statistic was used to compare the means. Each model was evaluated on the 6 held-out speed trials from training and an average loss was calculated for each model. The reference kinematics, externally driven and self-driven predictions of each of the 6 held-out trials per model were phase averaged and $R^2$ between the phase averaged externally driven and long-time self-driven predictions (*see Long-Time Performance section*, *above*) were calculated. Box plots for each metric across the held-out trials were generated and the Wilcoxon Rank-Sum Test statistic used to compare the means.

## Computing gait signatures from RNN internal states

To develop the gait signatures, we extract the activation and cell states from the LSTM (denoted "H" and "C" respectively) which evolve over time (the course of the gait cycle) as the kinematics of each trial are fed through the trained RNN. These H and C parameters represent how the model's internal parameters change as it encodes the prediction of future kinematic trajectories. The selected 512-node LSTM layer had 512 H parameters and 512 C parameters. Time-varying gait signatures were computed by identifying dominant modes of variation in the internal states using principal components analysis (PCA). A single PCA operation was used to transform the internal states for all participants into a common basis. Consequently,

inter-trial differences in the time-varying activations of the principal components (modes) reflect differences in the underlying dynamics of the individual(s). These activations constituted the time-varying (1500 sample) *gait signatures*, which had the same dimension as the RNN's hidden layer (1024 units). However, the first six principal components accounted for ~72% of the variance in the internal states.

To compare gait signatures within and between individuals, we phase averaged each trial's signatures across strides. Rather than linearly interpolating the data between foot contact events before averaging, as is common in gait analysis [122,125–127] we computed a continuous phase using the first 3 gait signature modes for each trial using Von Mises interpolation [128]. Compared to averaging across linearly interpolated strides, phase averaging is expected to reduce the variance in the data at any point in the stride [66,90]. As the domain for interpolation, we estimated the time-varying phase for each trial separately using the Phaser algorithm, using the first 3 principal components as phase variables [66]. To align phase estimates across trials, we defined zero-phase as the maximum of the first principal component.

### Gait event estimation of phase averaged signatures

The force plates embedded in the treadmill captured precise gait event timing information (left heel strike, right toe off, right heel strike, left toe off) across individuals' trials which we represented as a vector of 1's, 2's, 3's and 4's, respectively (ground truth markings for the 4 gait events). We leveraged the Phaser algorithm again [66] to develop a phase estimator to transform these 4 gait events over time into gait events over phase. For each trial, we determined the mode phase that corresponded to each of the 4 gait events to gain a representation of where the 4 gait events occurred during phase averaged dynamics ($0$–$2\pi$) for each trial.

### Interpolation of unseen speed gait signatures to reconstruct kinematics

To demonstrate the generalizability of gait dynamics, we show that linearly interpolating gait signatures to predict gait kinematics at new walking speeds is more accurate than linearly interpolating the kinematics themselves. We trained another RNN model with the same architecture and hyperparameters to the first model, however using only the 2 slowest and 2 fastest speeds from each participant (i.e., we held out the 2 middle speed trials from each participant). We then linearly interpolated the 2-middle speeds' internal states and ran the data through the trained RNN to reconstruct or predict kinematics. We compared the original phase averaged kinematics to the predicted kinematics resulting from each of the two linear interpolations (dynamics and kinematics) using the coefficient of determination. Furthermore, even when we reduced the dimensionality of the model's internal states from the full 1024 to the first 6 principal components (the selected dimension on the gait signatures), it still performed better than interpolating kinematics (also rank = 6).

### Biomechanical interpretation of principal components of the gait signature

To reconstruct kinematics from the corresponding underlying dynamics (internal state representations), we restored our trained model's weights to a new model using the 'model.set' and 'model.get_weights' Keras built in functions. The function 'model.predict' takes in the hidden state values (Hs) only (first 512 of the 1024 internal-state time trajectories) and predicts the corresponding kinematics for the provided internal states. Using this framework, we provided this new model with independent principal component representations of individuals' hidden states and visualized the corresponding kinematics through stick figure movie representations of the resulting kinematics over the walking trials.

## Predicting time evolution of kinematics from an initial posture (self-driving)

Our trained generative gait model can take in a single initial posture of size (6,1) corresponding to a single time point representation of each of the 6 joint angles to predict the next time step posture/kinematics using command 'model.predict'. To make further predictions, the predicted value is used as the new initial condition (posture) and predictions are made on a one-time step basis in a similar fashion for a pre-specified prediction length (self-driving). However, it is important to note that even though our current framework only predicts one time-step in the future, the LSTM layer remains stateful through each gait cycle, allowing the model to learn much longer timescales as seen when self-driving the network (S7 Fig).

## Supporting information

**S1 Video. Stick figure movie demonstrating the RNN's kinematic reconstruction of all the PCs of dynamics.**
(MP4)

**S2 Video. Stick figure movie demonstrating that PC1 encodes dynamics driving hip flexion and extension.**
(MP4)

**S3 Video. Stick figure movie demonstrating that PC2 encodes dynamics driving knee flexion and extension.**
(MP4)

**S4 Video. Stick figure movie demonstrating that PC3 encodes dynamics driving postural coordination.**
(MP4)

**S5 Video. Stick figure movie demonstrating the kinematic reconstruction when AB2's PC1 projection is replaced with ST4's impaired PC1 projection.**
(MP4)

**S6 Video. Stick figure movie demonstrating the kinematic reconstruction when AB2's PC3 projection is replaced with ST4's impaired PC3 projection.**
(MP4)

**S7 Video. Stick figure movie demonstrating the kinematic reconstruction when AB2's PC4 projection is replaced with ST2's impaired PC4 projection.**
(MP4)

**S8 Video. Stick figure movie demonstrating the kinematic reconstruction when ST3's impaired PC5 projection is replaced with AB2's PC5 projection.**
(MP4)

**S1 Fig. Comparison of model performance on training and validation loss (left), and long- and short-time prediction performance (right).** In both plots, small dots represent the average values across trials for each random initialization of each model. Large dots and bars denote the average and standard deviation of model performance metrics across initializations. Left: Training and validation loss (RMSE) for all 12 hyperparameter pairs. Models in the lower-left consider are considered better. Right: Long- and short-time prediction performance (R2) for the 3 hyperparameter pairs with the lowest training and validation loss. Models in the

upper-right corner are considered better.
(TIF)

**S2 Fig. RNN model training (green) and validation (blue) loss curves.**
(TIF)

**S3 Fig. RNN dynamic learning generalizes across 12 leave-one-individual-out models.** The minimum train loss (blue) and validation loss (green) was low ($< 0.02$ degrees2) for 5 models that were trained each with a single able-bodied individual held out of the training data (A, i) and 7 models each with a stroke individual held out of training data (A, ii). The magnitude and range of the test loss (evaluation of model on the held-out data) (orange) was higher than the respective minimum training and validation losses for both held-out able-bodied (A, i) and stroke (A, ii) models. The magnitude and range of test losses evaluated on held-out able-bodied individuals, however, were lower than models evaluated on held-out stroke data. The models generate external predictions (blue) of held-out test trials with higher R2 values than that of self-driven predictions (red) in models evaluated on both able-bodied (B, i) and stroke trials (B, ii). The models can generate external kinematic predictions (blue) of held-out able-bodied (B, i) trials better than that of stroke (B, ii). Self-driven predictions of stroke kinematics were generally very low (R2 values below 0.5). Models were incapable of generating self-driven predictions for 5 of 30 able-bodied trials and 16 of 42 stroke trials. (C) shows reference (black), externally driven (blue) and self-driven (red) phase averaged kinematic predictions for an exemplary able-bodied trial (C, i) and exemplary stroke trial (C, ii). Models can predict kinematics of held-out able-bodied trials better (higher R2) than that of stroke.
(TIF)

**S4 Fig. Cumulative proportion of variance explained by the first 100 principal components of gait dynamics.** Six (6) principal components (PCs) explained 77% of the variance in the gait dynamics. The top 6 dominant PCs were used to develop the gait signature.
(TIF)

**S5 Fig. Support vector machine cross-validation classification accuracy of four different gait descriptors (discrete variables, gait signatures, kinematics, and a combination of kinematics & joint velocity) for discrimination between: a) gait group (able-bodied vs. stroke) and B) individuals.** Using k = 25 folds, RNN gait signatures distinguished between impaired and unimpaired gait with 100% accuracy, along with the 26 discrete variables (100%, p = 1), whereas kinematic ($92.67 \pm 0.15\%$, $p < 0.05$) and kinematics & velocity ($88.67 \pm 0.17\%$, $p < 0.05$) discrimination were significantly lower. Using k = 6 folds, SVM classification of individuals was most accurate using RNN gait signatures and discrete variables (100%), lower using kinematics ($88.9 \pm 0.13\%$, p = 0.061) and significantly lower using a kinematics and velocity ($68.10 \pm 0.16\%$, $p < 0.05$).
(TIF)

**S6 Fig. Relationship between participants' similarity to able-bodied gait signatures and the RNN's ability to predict gait kinematics over one gait cycle.** There exists a negative correlation between self-driven R2 and Euclidean distance to the AB centroid that is statistically significant at the 0.05 level (S6 Fig, dots). Lower functioning stroke survivors are located further away from the able-bodied centroid, however remarkably all but one (outlier) of the R2 values are above 0.73 in both high and lower-functioning individuals. Even though the model can better predict able-bodied future kinematics better than stroke (R2 values above 0.8), the ability of our model to predict at least a single gait cycle of future stroke kinematics with R2 above 0.73 is promising. To provide a control for this analysis, we found the average R2 values

between the self-driven prediction of the last full gait cycle of each trial and 5 randomly selected gait cycles from other randomly selected individuals in the same gait group (able-bodied vs. stroke) (S6 Fig, stars). This control would effectively reveal whether our model learned each individual's specific gait pattern or was just producing arbitrary (averaged) gait patterns. We expect that if the model learned individuals specific gait patterns, then the distribution of the R2 values of the self-driven kinematic predictions would be significantly different to that of the averaged R2 values corresponding to arbitrary gait patterns. In fact, the averaged R2 of the generic gait cycle comparisons ranged from 0.56 to 0.95 (compared to 0.61–0.99 in self-driven predictions). Further, the distributions of the R2 outputs for both able-bodied and stroke individuals were significantly different (at the 0.05 level) to self-driven R2 outputs with p-values 5.5x10-6 and 5.4x10-3 respectively using the Wilcoxon Rank-Sum test. This reveals that even for stroke survivors the model has learned their dynamics in some capacity and the model isn't simply predicting arbitrary gait patterns.
(TIF)

**S7 Fig. Graphical summary and pseudo code of our gait signatures framework and algorithm. A) Training dataset:** Using the 6-dimensional gait trajectory, the inputs (green) were concurrent segments from the gait trajectory, each one 499-time steps long. The outputs (blue) were 1-shifted (in time) segments of inputs. **B) Model architecture:** Our model consisted of an input layer, a hidden layer composed of 512 LSTM units, and a 6-unit Dense output layer. **C) Stateful training:** The hidden state of an RNN at time t is a function of the input at time t and the hidden state at time t-1. The model starts with processing the first mini-batch, calculating a new hidden state at each t and predicting gait kinematics at time t+1 given kinematics data at t. At the end of the mini-batch processing, the model calculates MSE over the entire mini-batch to calculate error for backpropagation and to update model weights. The final hidden state h(t+L) is used as the initial hidden state for generating predictions for the next mini batch (L is the temporal length of each mini-batch). The hidden state is initialized as zero before processing the first mini-batch.
(TIF)

**S1 Table. Stroke participant's demographics and clinical scores.**
(TIF)

## Author Contributions

**Conceptualization:** Taniel S. Winner, Kanishk Jain, Trisha M. Kesar, Lena H. Ting, Gordon J. Berman.

**Data curation:** Trisha M. Kesar.

**Formal analysis:** Taniel S. Winner, Michael C. Rosenberg, Kanishk Jain, Trisha M. Kesar, Lena H. Ting, Gordon J. Berman.

**Funding acquisition:** Lena H. Ting.

**Methodology:** Taniel S. Winner, Kanishk Jain, Trisha M. Kesar, Lena H. Ting, Gordon J. Berman.

**Project administration:** Lena H. Ting.

**Resources:** Taniel S. Winner, Trisha M. Kesar, Lena H. Ting, Gordon J. Berman.

**Supervision:** Trisha M. Kesar, Lena H. Ting, Gordon J. Berman.

**Validation:** Taniel S. Winner, Michael C. Rosenberg.

**Visualization:** Taniel S. Winner, Michael C. Rosenberg.

**Writing – original draft:** Taniel S. Winner, Michael C. Rosenberg, Trisha M. Kesar, Lena H. Ting, Gordon J. Berman.

**Writing – review & editing:** Taniel S. Winner, Michael C. Rosenberg, Kanishk Jain, Trisha M. Kesar, Lena H. Ting, Gordon J. Berman.

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
