## [Decision Letter · Decision Letter 0]

14 Jul 2023

Dear Mrs. Winner,

Thank you very much for submitting your manuscript "Discovering individual-specific gait signatures from data- driven models of neuromechanical dynamics" for consideration at PLOS Computational Biology.

As with all papers reviewed by the journal, your manuscript was reviewed by members of the editorial board and by several independent reviewers. In light of the reviews (below this email), we would like to invite the resubmission of a significantly-revised version that takes into account the reviewers' comments.

We cannot make any decision about publication until we have seen the revised manuscript and your response to the reviewers' comments. Your revised manuscript is also likely to be sent to reviewers for further evaluation.

Sincerely,

Lyle Graham

Section Editor

PLOS Computational Biology

Reviewer's Responses to Questions

**Comments to the Authors:**

Reviewer #1: This is an exciting work on investigating and interpreting the neuromechanical dynamics accountable for distinct gait patterns using a machine learning framework, particularly recurrent neural networks (RNNs). The authors collected joint kinematics using motion capture from 12 adults (5 able-bodies and 7 stroke survivors) walking at different speeds (0.3-1.8m/s). The study found that the RNN model could detect individual gait signatures that were consistent across various walking speeds, with stroke survivors showing greater gait variations at slower speeds. Although the contributions of the work are significant and relevant to the clinical researchers in examining animal locomotion patterns, there are a few concerns, as follows, that question the technical soundness of the work.

1. Referring Figure 1(b) and 2(b), the authors claim that the cluster of low functioning stroke survivors at self selected speed<0.4m/s is at farthest distance from the able-bodied cluster. However, it is quite evident from the results that encapsulated cluster of low functioning stroke has several outliers, i.e., self selected speed>0.4 m/s (bigger circles). The authors are suggested to justify the reasoning behind ignoring the effect of such outliers.

2. Given Figure 5, the generalisation capability of the trained RNN model is checked by interpolating gait kinematics across speeds, taking 1.32 m/s and 0.43 m/s. Although both speeds are unseen,; however, within the slow and fast speed limit. The generalisation capability will be more evident if the unseen speed will be outside the training range.

3. It is quite obvious that the RNN model would able to generalise able body kinematics much better than stroke survivors due to the significant difference in impaired gait kinematics among the stroke survivors. In that case, given S3 Figure, two questions must be investigated, :

(a) How well the RNN model generalises for the high functioning stroke survivors? Is the self driven R^2 increasing due to closer cluster distance to centroid of able-bodied individuals?

(b) How one can establish a relation between the differences in joint kinematics (varying from high to low functioning stroke survivors) and impaired anthropometric parameters such as flexion/extension deformities.

4. Moreover, the authors are suggested to provide the medical details for lower-limb joints of the stroke patients. Have the authors considered existing discrepancies in lower-limb joints such as fixed flexion deformities (FFD)? Such discrepancies might affect the gait signatures significantly.

5. There should be a algorithm (pesudo code) to highlight the computational approach involved in the development and implementation of RNN model. Moreover, the authors are suggested to provide a schematic representation of the proposed architecture with hyperparameters. The authors should justify the reason behind the selection of ADAM optimizer. Moreover, the details of the optimiser should be highlighted to improve the readability of the paper.

6. (Line 212) The authors claim that the training and validation error converged for less than 0.75 deg. They are suggested to provide the error convergence plot in addition to S2 Figure.

7. (Line 246) The authors claim that six principal components (PCs) explained 77% of the variance in the gait dynamics. However, in view of S4 Figure, six PCs are able to explain ~71% of the gait variance. The authors are suggested to justify the reason of this discrepancy in the text.

8. This study have focused on a specific gait pathologies (cortical or subcortical ischemic stroke) and only for seven patients, which may limit the applicability of the findings to other gait pathologies. This limitation can not be ignored by just mentioning in the discussion. Additionally, the study have not accounted for confounding variables, such as medical conditions or environmental factors, which could affect gait patterns.

Reviewer #2: The paper proposes a data-driven approach to analyze individual gait dynamics of able bodied individuals and of stroke survivors. The model is based on a simple one-layer LSTM network, used to predict the evolution of the pose of the subjects during locomotion at different speeds. The hidden state of the LSTM is used to create a reduced base on which phase-averaged LSTM activations can be projected. Such projection, defined "locomotion signature", is analyzed in the attempt to identify differences in the gait across subjects. The main findings are:

- Able bodied, high-functioning and low-functioning stroke survivors more or less cluster together

- Able bodied gaits form a more packed cluster

- The variability across gaits of different individuals exceeds that within an individual (even at different speeds), justifying the definition of "locomotion signature"

- Individual components of the locomotion signature are interpretable. Parts of the signature from different individuals can be combined, to investigate specific locomotion deficits.

- The LSTM can be used as a generative model. The able bodied locomotion can be generated accurately over many cycles, while that of stroke survivors is only accurate for one cycle, and then converges to that of able bodied individuals.

In my opinion the paper is very well written and motivated. The exposition is clear and precise. The methods are adequate and the authors have good knowledge of the tools they use.

I have some questions and comments:

- I am surprised that the differences in the gait at different locomotion speeds are much smaller than the differences between the gait of different individuals. I speculate that this is caused by 2 facts: 1. the sample size is small. With 13 individuals I can imagine that each gait has some unique, distinguishable features. If they had considered, e.g., 1000 individuals, I am quite sure the differences would have not been visible. Therefore, does it make sense to speak about a signature, when we do not know whether individual gaits would emerge in a larger dataset? 2. I think the speed range is also small. The authors hint to this fact (lines 289-290). This is a strong limitation, as one of the main results is that the variability across individuals exceeds the variability across locomotion speeds. The full range of locomotion speeds should be considered to prove something like this.

- The gait signature alignment is part of the model selection. Maybe it does not have a strong impact, but I am not sure it is correct to include the point you want to prove (that the hidden state of the LSTM defines a locomotion signature) it the selection of the model to use?

- It is briefly mentioned that gait signatures perform similarly to 26 commonly used discrete variables (fig. S5). The authors should better clarify why the features derived from the LSTM state should be preferred to these variables. They perform identically in classifying gait groups and individuals.

- There are some further clusters in the plots: there seem to be 2 able bodied clusters, almost as far from each other as the stroke survivors. What are these differences due to?

- Fig. 3: not all low-functioning subclusters really look further from the able-bodied than the high-functioning ones. Overall, I doubt that any clustering algorithm would find the able-bodied, high-funcitoning and low-functioning clusters

- Retraining the LSTM on any different dataset (or even just with different initialization) would lead a different basis vector, meaning that the analysis of the PCs has to be done for every model. Do they propose that the basis they found in this paper is used also in future studies? Or how do we do in practice?

- The analysis named "gait sculpting" is very original and interesting in my opinion

- the section 547-557 is not fully clear to me. Have the authors tried to set as an objective of the LSTM a longer trajectory, instead of the next pose? This might improve a lot the generative power of the network.

Minor comments:

104 missing comma?

297 missing )

299 "similarly to a when"?

572-573 strange sentence

660-662 not clear that this result actually follows from the study

737 strong title given the quality of the future predictions

752 extra full stop

**Have the authors made all data and (if applicable) computational code underlying the findings in their manuscript fully available?**

Reviewer #1: Yes

Reviewer #2: Yes

PLOS authors have the option to publish the peer review history of their article (what does this mean?). If published, this will include your full peer review and any attached files.

Reviewer #1: No

Reviewer #2: No
---

## [Decision Letter · Decision Letter 1]

30 Sep 2023

Dear Mrs. Winner,

We are pleased to inform you that your manuscript 'Discovering individual-specific gait signatures from data- driven models of neuromechanical dynamics' has been provisionally accepted for publication in PLOS Computational Biology.

I also suggest that you take into account the suggestion from Reviewer 2 about the Discussion, which seems quite reasonable.

Best regards,

Lyle Graham

Section Editor

PLOS Computational Biology

Reviewer's Responses to Questions

**Comments to the Authors:**

Reviewer #1: The authors have addressed all the concerns raised by the reviewer. The manuscript is now in better shape and organization with improved quality. In future, it would be interesting to see the extended work from the authors' group on using diverse dataset and learning algorithm with better generalization (extrapolating) capabilities.

Reviewer #2: We thank the authors for their detailed response. Overall, we think the paper (was already good) and improved further. As is clear from many responses, this framework would shine most, and perhaps provide interesting insights with a larger cohort of people. E.g., from your answer, I gather that you only test walking speeds from 0 to 1.6 m/s, which is far away from running speed. They do cover the whole range of safe speeds for stroke survivors, but definitely not for able bodied patients. Thus, I do think this limitation should be clarified in the discussion. That said, the dataset is not too small to draw the conclusions.

**Have the authors made all data and (if applicable) computational code underlying the findings in their manuscript fully available?**

Reviewer #1: Yes

Reviewer #2: Yes

PLOS authors have the option to publish the peer review history of their article (what does this mean?). If published, this will include your full peer review and any attached files.

Reviewer #1: **Yes: **Jyotindra Narayan

Reviewer #2: No

---

## [Editor Report · Acceptance letter]

13 Oct 2023

PCOMPBIOL-D-23-00152R1 

Discovering individual-specific gait signatures from data-driven models of neuromechanical dynamics

Dear Dr Winner,

I am pleased to inform you that your manuscript has been formally accepted for publication in PLOS Computational Biology. Your manuscript is now with our production department and you will be notified of the publication date in due course.

With kind regards,

Anita Estes
